# Spatial cue reliability drives frequency tuning in the barn Owl's midbrain

Fanny Cazettes[1]*, Brian J Fischer[2]†, Jose L Pena[1]†

[1]Department of Neuroscience, Albert Einstein College of Medicine, New York, United States; [2]Department of Mathematics, Seattle University, Seattle, United States

**Abstract** The robust representation of the environment from unreliable sensory cues is vital for the efficient function of the brain. However, how the neural processing captures the most reliable cues is unknown. The interaural time difference (ITD) is the primary cue to localize sound in horizontal space. ITD is encoded in the firing rate of neurons that detect interaural phase difference (IPD). Due to the filtering effect of the head, IPD for a given location varies depending on the environmental context. We found that, in barn owls, at each location there is a frequency range where the head filtering yields the most reliable IPDs across contexts. Remarkably, the frequency tuning of space-specific neurons in the owl's midbrain varies with their preferred sound location, matching the range that carries the most reliable IPD. Thus, frequency tuning in the owl's space-specific neurons reflects a higher-order feature of the code that captures cue reliability.

## Introduction

Perception relies on sensory cues that are used by the brain to infer properties of the environment. For example, the ability to see the world in three dimensions depends on cues that signal depth (*Howard, 2012*). Similarly, sound localization, relies on auditory spatial cues including phase differences of sounds between the ears (*Moiseff and Konishi, 1983*; *Grothe et al., 2010*). Variability of sensory cues is intrinsic to the physics of stimuli and sensory organs. For instance, light and sound waves are reflected and absorbed differently by various media depending on wavelength and location (*Carlile, 1996*). Multiple light and sound sources can also physically interfere with each other. In the auditory system, contexts such as whether the environment is reverberant, quiet or noisy can influence spatial cues greatly. The presence of concurrent sounds can shift auditory cues used for localizing a target sound (*Keller and Takahashi, 2005*). This shift makes cues differ from what would be measured if the sound was emitted in a quiet environment. To be considered reliable, cues associated with a given location must be similar across different contexts. Unreliable cues, on the other hand, vary across contexts.

The brain must take into account the reliability of sensory cues in order to make perception robust to natural variations. A possible strategy to overcome these variations is to integrate sensory cues with reference to their variability. Indeed, when cues provide ambiguous or conflicting information, those cues that vary less are weighted more heavily in perceptual judgments (*Ernst and Banks, 2002*; *Jacobs, 2002*; *Alais and Burr, 2004*).

Humans and other animals use the interaural time difference (ITD) for sound localization (*Grothe et al., 2010*). ITD is the difference in the arrival time of a sound at the ears. ITD results from unequal distances of a sound source to the two ears when the source is to the left or to the right of the listener. In barn owls, ITD is the main cue for localizing in the horizontal space (*Moiseff and Konishi, 1983*; *Moiseff, 1989*). ITD is initially detected by brainstem neurons tuned to narrow frequency bands in both mammals and birds (*Carr and Konishi, 1990*; *Schnupp and Carr, 2009*; *Figure 1A*).

*For correspondence: fanny. cazettes@phd.einstein.yu.edu

†These authors contributed equally to this work

Competing interests: The authors declare that no competing interests exist.

**eLife digest** The ability to locate where a sound is coming from is an essential survival skill for both prey and predator species. A major cue used by the brain to infer the sound's location is the difference in arrival time of the sound at the left and right ears; for example, a sound coming from the left side will reach the left ear before the right ear.

We are exposed to a variety of sounds of different intensities (loud or soft), and pitch (high or low) emitted from many different directions. The cacophony that surrounds us makes it a challenge to detect where individual sounds come from because other sounds from different directions corrupt the signals coming from the target. This background noise can profoundly affect the reliability of the sensory cue.

When sounds reach the ears, the head and external ears transform the sound in a direction-dependent manner so that some pitches are amplified more than other pitches for specific directions. However, the consequence of this filtering is that the directional information about a sound may be altered. For example, if two sounds of a similar pitch but from different locations are heard at the same time, they will add up at the ears and change the directional information. The group of neurons that respond to that range of pitches will be activated by both sounds so they cannot provide reliable information about the direction of the individual sounds. The degree to which the directional information is altered depends on the pitch that is being detected by the neurons; therefore detection of a different pitch within the sound may be a more reliable cue.

Cazettes et al. used the known filtering properties of the owl's head to predict the reliability of the timing cue for sounds coming from different directions in a noisy environment. This analysis showed that for each direction, there was a range of pitches that carried the most reliable cues. The study then focused on whether the neurons that represent hearing space in the owl's brain were sensitive to this range.

The experiments found a remarkable correlation between the pitch preferred by each neuron and the range that carried the most reliable cue for each direction. This finding challenges the common view of sensory neurons as simple processors by showing that they are also selective to high-order properties relating to the reliability of the cue.

Besides selecting the cues that are likely to be the most reliable, the brain must capture changes in the reliability of the sensory cues. In addition, this reliability must be incorporated into the information carried by neurons and used when deciding how best to act in uncertain situations. Future research will be required to unravel how the brain does this.

These neurons compare the timing of inputs from the left and right sides of the brain (*Carr and Konishi, 1990*). Due to the periodicity of sound signals when narrow frequency channels are considered, the shift in time between the left and right ears for each frequency component is more precisely expressed in terms of phase, referred to as the interaural phase difference (IPD). The spectrum of IPDs across frequency serves as a set of cues used for sound localization.

Previous studies of sound localization suggest that the emergence of space-specific neurons in the midbrain is due to a wide convergence of frequency channels (*Brainard et al., 1992*, *Mazer, 1998*, *Saberi et al., 1998*; *Figure 1A*). These studies focused on how the brain infers sound location from IPD without taking into account that IPD at a given sound location may vary between contexts in a frequency-dependent manner. In fact, a direct consequence of the filtering effect of the head is that the IPD induced by different acoustic objects varies in each frequency channel depending on context (*Blauert, 1997*; *Keller and Takahashi, 2005*). For example, the phase of a target sound can be shifted by the presence of a concurrent sound and the amount of phase shift is frequency dependent (*Keller and Takahashi, 2005*; *Figure 1B*). Thus, for a given location of a sound source, the IPD cue may vary in a frequency-dependent manner in the presence of another sound. In other words, IPD may not be equally reliable as a sound localization cue at every frequency band. Whether the neural code for sound localization captures the reliability of auditory cues across contexts is unknown.

A common strategy for dealing with the natural variation of sensory cues is to weight cues in proportion to their reliability (*Ernst and Banks, 2002*; *Jacobs, 2002*; *Alais and Burr, 2004*; *Fetsch et al., 2012*). For sound localization based on IPD, this would mean that those frequencies that elicit more

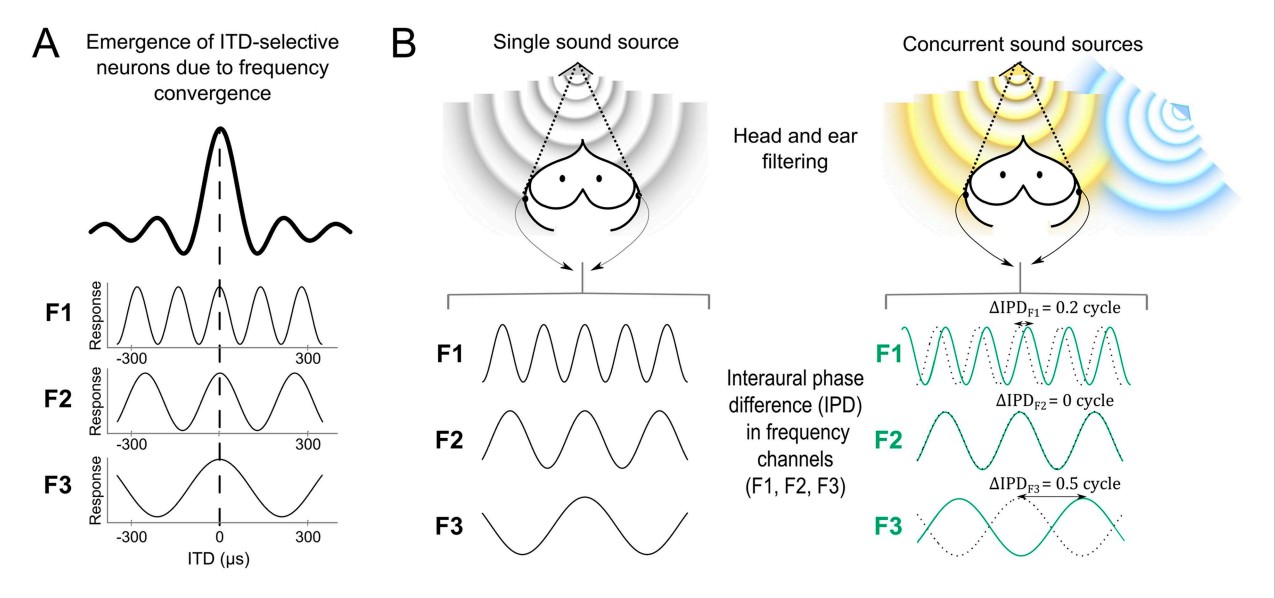

**Figure 1**. Cue variability in the sound localization system. (**A**) Tuning to interaural time difference (ITD) emerges from the convergence of inputs selective for the same ITD but different frequencies (F1–F3) (*Takahashi and Konishi, 1986*; *Mazer, 1998*; *Pena and Konishi, 2000*; best ITD indicated by the dashed line). While ITD-selective neurons respond to a broad range of frequencies (here the black bold curve represents the neuron's combined response for frequencies F1, F2 and F3), their inputs are narrowly tuned to frequency (each input only responding to F1, F2 or F3). Because the inputs are narrowly tuned to frequency, the responses at each input vary with the phase difference between the left and right ears (IPD) of their preferred frequency, as shown by the sinusoidal curves. (**B**) A sound emitted by a single source (**B**, left) in front of the owl is filtered by the head and decomposed in narrow frequency channels by the cochlea. The localization cue corresponds to an IPD in each frequency channel (F1–F3). In a different context (**B**, right), the target frontal sound (yellow) is emitted concurrently with another sound source from a different location (blue). The blue source interferes with the yellow target and shifts the resultant IPDs in each frequency channel (shown in green). Black dotted lines indicate IPD responses for the target frontal source alone, for comparison. While IPD shifts greatly for some frequencies (F1 and F3), in others (F2) IPD is more robust to the presence of another source. Thus in this example, F2 carries the most reliable IPD cue. To provide a clearer visualization that IPD is encoded at different frequencies, F1–F3 inputs remain plotted as a function of ITD in (**B**).

reliable IPD (i.e., where the variability of IPD is small) at a given location across different contexts should be favored in the process of estimating this particular location. We will refer to this mechanism as *weighting by reliability*. To test whether weighting by reliability occurs in sound localization, one can examine populations of neurons that integrate localization cues. A neural representation of auditory space emerges in the barn owl's external nucleus of the inferior colliculus (ICx). Thus, the barn owl's ICx provides an opportunity to test whether weighting by reliability is used to map sound location from variable cues.

In the present study, we demonstrate that frequency tuning of ICx neurons changes with their preferred ITD. This coding captures the variability of IPD across frequency channels in a manner consistent with weighting neural responses by cue reliability.

## Results

### Tuning to ITD and frequency

To test whether weighting by reliability occurs in the owl's sound localization pathway, we first mapped the spatial and frequency tunings of the entire ICx. 177 single units obtained from 138 recording sites in the ICx of two adult barn owls were included in this analysis. Single unit recordings were validated by spike sorting software (*Quiroga et al., 2004*) in 99 of the 138 recordings sites, whereas in 39 recording sites spike sorting separated two different units. Therefore, while the majority of recordings consisted of single units, some of them yielded two units. ITD and frequency tuning were measured for each neuron. We estimated best ITD from the peak of the rate-ITD tuning curve (*Figure 2A*, top row) and the best frequency (BF) from the center of the rate-frequency tuning curve (*Figure 2A*, bottom row). The neurons' best ITD is correlated with the preferred azimuth in the map of auditory space of the owl's ICx (*Moiseff and Konishi, 1983*). To achieve a representative

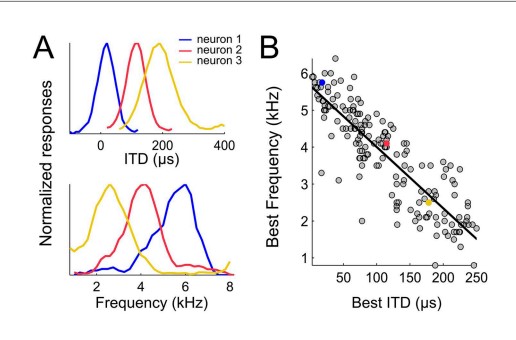

**Figure 2**. Spatial-dependence of frequency tuning in the population of ICx neurons. (**A**; top) ITD tuning measured with broadband noise of three example neurons tuned to 0 μs (blue), 100 μs (red) and 200 μs (yellow). (**A**; bottom) The frequency tuning, measured with tones at the best ITD shows that best frequencies (BF) decrease as best ITD increases for each neuron (BFs of the shown examples are 6 kHz (blue), 4 kHz (red) and 2 kHz (yellow)). (**B**) BF decreases with best ITD across the sample of ICx neurons. Linear regression is indicated by a solid line.

assessment of the neural population, we recorded responses of ICx neurons with best ITDs spread over the entire map. Best ITD varied from 0 to 249 μs and BF varied from 920 Hz to 6168 Hz. To the best of our knowledge, neurons tuned to such large ITDs or low BFs have not been reported in the owl's ICx (*Perez and Pena, 2006*; *Wagner et al., 2007*; *Vonderschen and Wagner, 2009*). As shown in *Figure 2*, best ITD was tightly correlated with BF ($r^2 = 0.75$, $p < 0.001$). Neurons preferring small ITDs were tuned to higher frequencies and, conversely, those preferring large ITDs were tuned to lower frequencies. The strong correlation between best ITD and BF was not affected whether the analysis was performed using recording sites with a single unit (n = 99, $r^2 = 0.71$, $p < 0.001$), pooled recordings of single and multi-units (n = 138, $r^2 = 0.72$, $p < 0.001$) or all sorted single units (n = 177, $r^2 = 0.75$, $p < 0.001$). All further analyses were thus performed on sorted single-unit data.

## IPD reliability

To investigate IPD reliability, we examined the statistics of the auditory input for the owl. Specifically, we considered how IPD varies when concurrent sources are present. The presence of concurrent sounds is likely a primary source of variability of IPD in the owl's natural environment because sounds made by prey are often faint (*Konishi, 1973*) and concurrent sounds from different locations can dramatically shift the measured IPD (*Keller and Takahashi, 2005*).

In the owl, and other species, sounds are modified in a location-dependent manner by facial structures. In barn owls, which lack a pinna, the facial ruff and ear canal act as filters (*Keller et al., 1998*; *von Campenhausen and Wagner, 2006*) embodied in the head-related transfer functions (HRTFs; *Keller et al., 1998*). These filters change both the phase and the magnitude of each frequency component of the sound in a location-dependent manner. When sounds are coming from multiple sources, the sound waves from each source will add in the ears and alter binaural cues. As described in *Keller and Takahashi (2005)*, the binaural cues resulting from the mixture of multiple sound sources are dictated by the relative intensity of each source within each frequency band. If two sources emitting sounds at the same intensity differ in the IPD within a frequency band, the resultant IPD is the average of the IPDs from the individual sources. When a target source carries more power in a particular frequency band, the resulting IPD will shift closer to this source. If the power of the second source is larger at a given frequency band, the IPD at this band is drawn away from the target source. *Figure 3A* shows an example of the relative variation in IPD around the mean IPD caused by the presence of concurrent sources. In this example the IPDs generated by one source located at 5° of azimuth and 0° of elevation (black dots) is contrasted with IPDs obtained when a second source, also at 0° elevation, is added at azimuths ranging between −90 and 90° (grey dots). For this azimuthal location, IPD variability induced by a second source is greatest between 3 and 4 kHz.

We measured IPD variability induced by the presence of concurrent sound sources using the HRTFs of 10 owls. We computed IPD variability on a frequency-by-frequency basis as the standard deviation of IPD (expressed in percent of cycle) over different locations of the second source (within ±90deg). The presence of a concurrent sound had a powerful effect on IPD, accounting for variations of up to 30% of a cycle. The IPD variability was highest for locations in the periphery, especially at high frequencies (*Figure 3B*). The presence of two sources also increased IPD variability at low frequencies for target locations near the center (*Figure 3A,B*).

This pattern of IPD variability with concurrent sounds can be explained by the direction and frequency dependence of the intensity gain of the HRTFs. The intensity gain is largest for azimuths near the front and decreases significantly for eccentric locations (*Keller et al., 1998*). Therefore, the IPD of

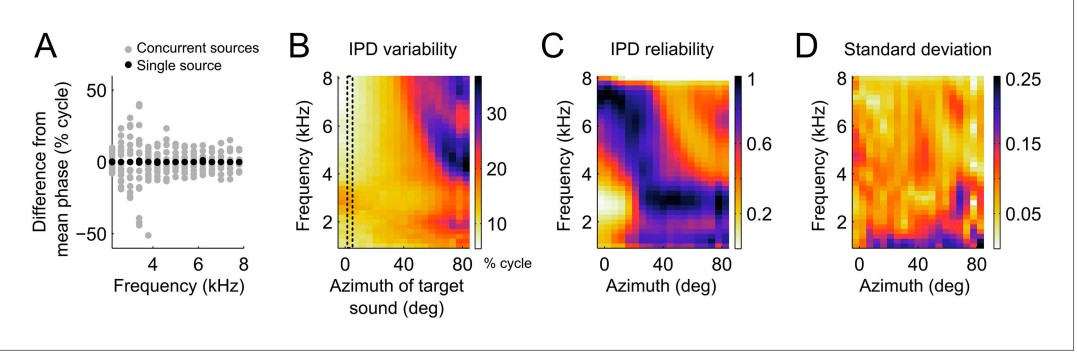

**Figure 3**. IPD variability computed from the head filters. (**A**) Example IPD variation around the mean (normalized to zero for clarity) as a function of frequency when a sound is presented near the front (black dots) and when a second sound source is added from various locations between −90 and 90° (gray dots). Note the larger scatter of gray dots in the lower frequencies (**B**) IPD standard deviation over concurrent sound sources across frequency at the location of the target sound. The dotted lines show the location of the target sound in (**A**). In (**A**) and (**B**) units are percent of cycle. (**C**) IPD reliability across location and frequency, normalized by the maximum at each location. (**D**) Standard deviation of the IPD reliability across HRTFs from 10 different owls. Note that the same colors map the values 0 (least reliable IPD) to 1 (most reliable IPD) in **C** and 0–0.25 in **D**.

a target source placed near the center will not be shifted greatly when a second source is placed at eccentric directions. Conversely, the IPD of a target source in the periphery will be shifted significantly when a second source is placed at a central location with high intensity gain. The difference in gain between frontal and peripheral locations is highest at high frequencies. Therefore, the largest variability was observed for high frequencies in the periphery.

We then took the inverse of the IPD variance as a measure of the reliability of IPD (*Figure 3C*). We assumed that an ICx neuron with a given preferred location will weight the inputs from different frequencies according to their relative reliability at that location. We therefore normalized the reliability within each location. We found that IPD reliability depends on the filtering of sounds by the head in a systematic manner across frequency and locations. Overall, IPD reliability was greater at high frequencies for locations in the front and at lower frequencies for locations in the periphery (*Figure 3C*). The overall pattern of IPD reliability was consistent across HRTFs from 10 different owls as illustrated by the small variance across animals (*Figure 3D*). If the frequency tuning of ICx neurons were driven by cue reliability, then we would expect a strong relationship between frequency tuning and IPD reliability.

## Testing the weighting by reliability hypothesis

The dependence of BF on ITD tuning (*Figure 2*) was predicted by the IPD reliability. Across neurons, their BF and the frequency at which IPD reliability was maximal at their preferred location were highly correlated ($r^2 = 0.81$, $p < 0.001$). Additionally, we compared the lower- and upper- edges of the neurons' frequency tuning curves with the range of frequency that carried the most reliable IPD (see 'Materials and methods'). The upper- and lower-frequency edges of the frequency tuning curves followed the upper ($r^2 = 0.72$, $p < 0.001$) and lower edges ($r^2 = 0.52$, $p < 0.001$) of the range of most reliable IPD (*Figure 4A*). Thus, the measured frequency tuning fell within the boundaries predicted by the IPD reliability. Since the match between frequency tuning curves and IPD reliability was assessed for each location, the normalization of the IPD reliability had no effect on these calculations.

To verify that additional sound sources did not change the pattern of IPD variability, we computed the IPD reliability with three concurrent sounds. The measured BF and the center frequency at which IPD reliability from three concurrent sounds was maximal were also highly correlated ($r^2 = 0.77$, $p < 0.001$).

ICx neurons are highly sensitive to interaural correlation, defined as the cross-correlation of the signals at the two ears (*Albeck and Konishi, 1995*). The addition of a concurrent source may decrease the interaural correlation at the preferred location of ICx neurons. Thus, in addition to the shift in IPD, the presence of concurrent sounds could modulate the firing rate of ICx neurons by changing interaural correlation. To test whether the IPD reliability is consistent with the pattern of interaural correlation induced by concurrent sounds, we examined the mean interaural correlation at the ITD of the target

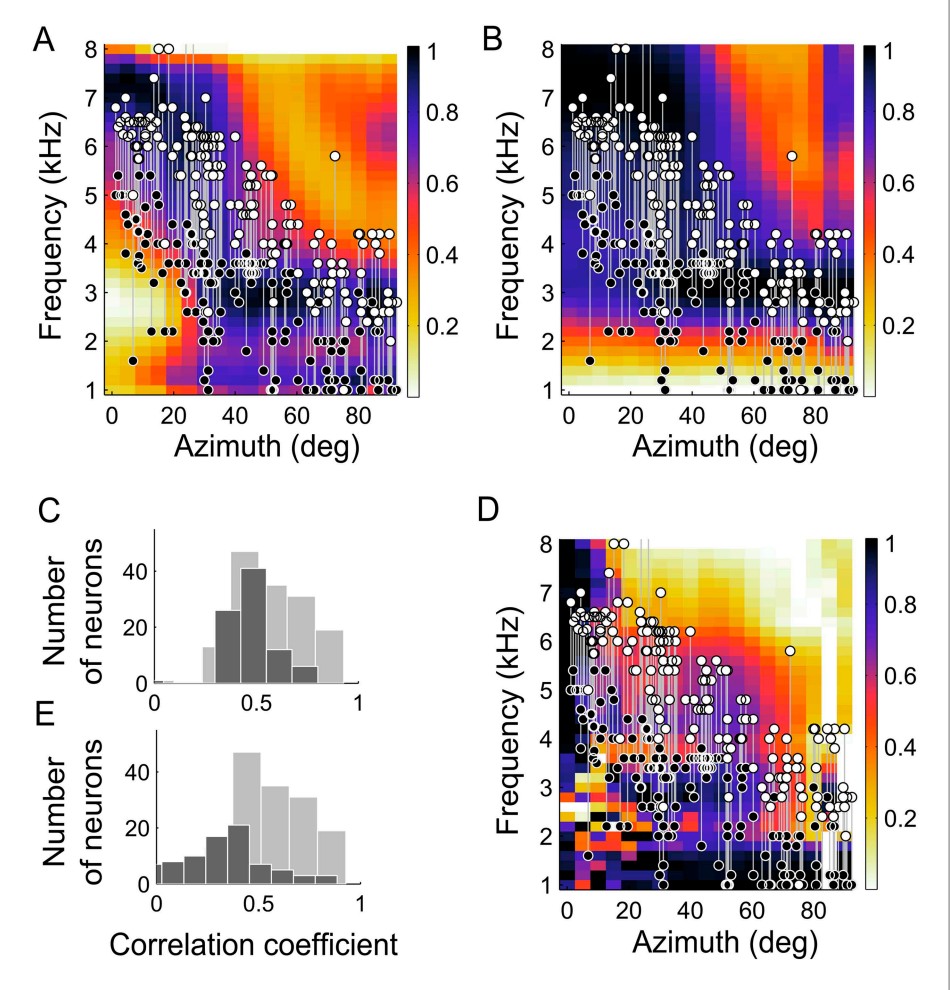

**Figure 4**. Frequency tuning in ICx matches IPD reliability. (**A**) Upper (white circles) and lower (black circles) edges of frequency tuning of ICx neurons superimposed on the plot of IPD-reliability. (**B**) Upper and lower boundaries of frequency tuning superimposed on the average gain normalized at each location (from 0 to 1). (**C**) The correlation coefficients between frequency tuning curves and frequency tuning predicted by the IPD-reliability (gray) are higher than the correlation coefficients between frequency tuning curves and gain alone (black). (**D**) Upper and lower edges of frequency tuning superimposed on the average gain normalized successively at each frequency and at each location (from 0 to 1). (**E**) The correlation coefficients between frequency tuning curves and frequency tuning predicted by the IPD-reliability (gray) are higher than the correlation coefficients between frequency tuning curves and the normalized gain (black).

source within each frequency channel while concurrent sounds from other locations were presented. We found that the pattern of interaural correlation as a function of frequency and location was very similar to the IPD reliability ($r^2$ = 0.81, $p$ < 0.001). Like IPD reliability, the mean interaural correlation was greater at high frequencies for locations in the front and at lower frequencies for locations in the periphery. The measured BF and the center frequency at which the mean interaural correlation was maximal were also highly correlated ($r^2$ = 0.88, $p$ < 0.001). This shows that weighting frequencies by their reliability maintains frequencies with high input coherence to ICx neurons at each location.

## Testing alternative hypotheses for the correlation between BF and best ITD

The change in sound-level due to filtering by the head, referred to as gain, is also frequency- and location-dependent (*Keller et al., 1998*; *Figure 4B*) and could drive frequency tuning. To investigate whether reliability of IPD or simply sound level was more important in driving the tuning of ICx

neurons, we asked whether frequency tuning was better predicted by IPD reliability (*Figure 4A*) or by the gain (*Figure 4B*). Gain and IPD reliability patterns displayed similarities, as expected from the fundamental relationship between the interaction of concurrent sound sources and their gain (*Keller and Takahashi, 2005*); IPD at locations with lower gain is more susceptible to variation in the presence of other sounds than is IPD at locations with higher gain. Additionally, correlation between IPD reliability and gain may also arise from common causes of variation such as acoustic reflections at the facial ruff (*von Campenhausen and Wagner, 2006*). However, the frequency tuning of neurons did not match the gain pattern as well as it matched the IPD reliability as a function of frequency at the best ITD (*Figure 4A,B*). We computed the correlation coefficients between the frequency tuning curves and both the IPD reliability and gain as a function of frequency at the best ITD of each neuron. Out of 177 neurons, the frequency tuning curves of 145 neurons (82%) were significantly correlated with IPD reliability as a function of frequency at the best ITD (*Figure 4C*, gray histogram) whereas only 85 neurons (48%) had their frequency tuning curves significantly correlated with the gain as a function of frequency at the best ITD (*Figure 4C*, black histogram). Further, the correlation between frequency tuning curves and IPD reliability yielded higher correlation coefficients than between frequency tuning and gain (p < 0.001, Wilcoxon rank-sum test).

It has also been shown that the relative gain of different frequency bands can be a cue for sound localization (*Butler, 1987*; *Butler and Musicant, 1993*). A critical feature of the owl's facial ruff is its ability to increase the intensity of the sound of high over low frequencies in the frontal space (*Keller et al., 1998*). At low frequencies, the owl's facial ruff has a relatively small location-dependent effect on sound level (*Hausmann et al., 2010*). Thus, the relative gain of high and low frequencies at different locations could provide a cue for stimulus location. To test whether the relative gain of each frequency could predict the ITD tuning, we normalized the gain across locations for each frequency separately before normalizing the gain across frequency for each location (*Figure 4D*). Once again, the results for the gain did not correlate with the experimental frequency tuning as well as the IPD reliability did (*Figure 4D,E*, p < 0.001 Wilcoxon rank-sum test). Only 44 neurons (25%) were significantly correlated with the normalized gain (*Figure 4E*, black histogram). Thus IPD reliability yielded a better prediction of the frequency tuning than did the gain.

To test whether changes in neural responses when concurrent sounds are present were consistent with predictions made from the HRTFs, we examined spatial tuning in neurons of the core of the central nucleus of the inferior colliculus (ICCc). ICCc is located earlier in the pathway leading to ICx and contains ITD-sensitive neurons that are narrowly tuned to frequency (*Wagner et al., 2002*). While ITD tuning varies with frequency in ICx, such dependence is not observed in ICCc (*Wagner et al., 2002, 2007*). Thus, recording in ICCc allowed us to assess sensory-input variability before frequency convergence occurs. Because the predictions due to reliability or to gain differed most between high and low frequencies in frontal locations (*Figure 4A,B,D*), we explored this range in ICCc using concurrent sounds. We measured the spatial tuning of ICCc cells using single sources and while another sound was presented from different locations covering the frontal hemisphere on the horizontal plane (*Figure 5A*, see 'Materials and methods'). To quantify the effect of concurrent sounds in altering the spatial tuning of ICCc cells, we cross-correlated the spatial tuning curves obtained with single sources with those measured with concurrent sounds (*Figure 5B*). We found that the spatial tuning of ICCc cells tuned to the front (preferred location between 5 and 20°) was more affected by the presence of another sound, that is, more variable, at lower than at higher frequencies (*Figure 5A,B*, r² = 0.6, p < 0.001). Experiments were performed in a high quality anechoic chamber (see 'Materials and methods'), thus we consider it unlikely that the variability in spatial tuning of low frequency ICCc neurons is due to room reflections. Therefore the data from ICCc neurons confirm our prediction that the variability induced by the HRTFs is carried over the sound localization pathway.

It has been proposed that the interaural canal connecting the middle ear cavities in birds could affect ITD at low frequencies (*Calford and Piddington, 1988*). Because HRTFs are measured using microphones positioned at the ear canal, they are blind to the effect of the interaural canal. The largest ITD predicted by the HRTFs at low frequencies was similar to the largest best ITD we recorded in ICx. Therefore it does not appear that the interaural canal has an important effect increasing the magnitude of ITD. However, if the interaural canal increased the gain of low frequencies dramatically in the frontal space, it could change the variability induced by concurrent sound sources. Yet, our results in ICCc are consistent with predictions made from the HRTFs without the need to invoke an effect of the interaural canal.

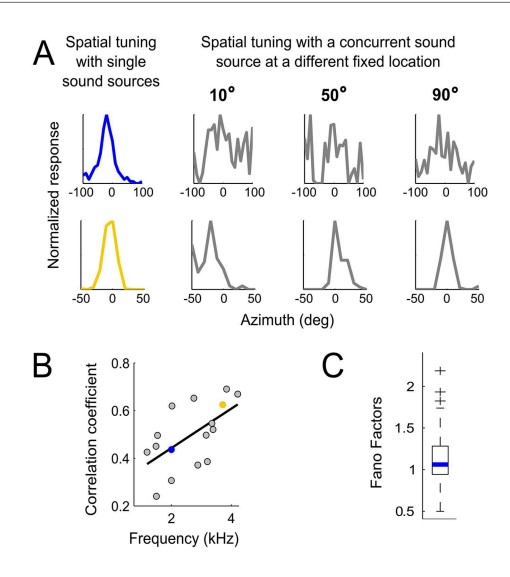

**Figure 5**. Testing the weighting by reliability hypothesis. (**A**) The colored curves show the normalized spatial tuning of two example ICCc neurons whose best frequencies are 1 kHz (blue) and 4 kHz (yellow). The spatial tuning was measured by varying the location of single sound sources using an array of speakers. The gray curves show the normalized spatial tuning of the same neurons measured with an additional concurrent sound at another location (nine different locations of concurrent sounds were tested in total). Examples of the spatial tuning measured in the presence of a concurrent sound source at 10°, 50° and 90° are displayed. The tuning of the low frequency neuron (top row) is more affected by concurrent sounds than the tuning of the higher frequency neuron (bottom row). (**B**) For each neuron, we correlated the curves measured with single sound sources with the curves measured with concurrent sound sources. The mean correlation coefficients between spatial tuning curves using a single sound and the tuning curves using concurrent sounds increases with best frequency. The linear regression is indicated by a solid line. The similarity between the curves in different contexts increases with best frequency. (**C**) Box-plot showing median (blue line) and quartiles of the Fano factor distribution. The Fano factors spread around 1, indicating trial-to-trial variability similar to a Poisson response.

There are a number of possible sources of neural noise that may contribute to the frequency dependence of IPD variability. While neural noise would not affect our measurements of BF or best ITD, which rely on tuning functions using mean firing rate, frequency-dependent neural noise during development could influence the learned connectivity in the auditory system that establishes the correlation between BF and best ITD. To assess the frequency dependence of neural noise we computed the Fano factors, a measure of trial-to-trial response variability, for ITD tuning curves obtained with tonal stimulation. We measured ITD tuning curves using tones from 1 to 7 kHz in 70 ICx cells (total of 342 curves). If neural noise were correlated with frequency, Fano factors should vary with stimulus frequency. We found that the Fano factors spread around 1 (*Figure 5C*, median = 1.06) and were negligibly correlated with the stimulus frequency ($r^2 = 0.1$, $p = 0.03$). We also assessed whether neural noise is frequency-dependent when examined at different locations. We split the neurons into 4 groups according to their best ITDs (0–50 µs; 50–100 µs; 100–150 µs; larger than 150 µs). For each group we averaged the Fano factors of the different neurons across the same stimulating frequency. We found no relationship between the Fano factor and frequency as a function of best ITD in the first 3 groups (Group1: $r^2 = -0.08$, $p = 0.68$; Group2: $r^2 = 0.007$, $p = 0.92$; Group3: $r^2 = 0.1$, $p = 0.62$). For best ITDs larger than 150 µs, the Fano factor decreased as frequency increased ($r^2 = -0.7$, $p < 0.001$). However, if neural noise drove the low-pass frequency tuning at eccentric locations, the opposite relationship would be expected (higher frequencies noisier than lower frequencies). Thus, frequency-dependent trial-to-trial variability in neural responses cannot explain the correlation between best ITD and frequency tuning.

Phase locking often weakens at higher frequencies, thus potentially increasing IPD variability at these frequencies (*Koppl, 1997b*). To test this, we examined whether the strength of IPD tuning varied with the stimulation frequency using a synchronization coefficient (*Goldberg and Brown,*

*1969*; *Kuwada et al., 1987*). In the same dataset used to assess the Fano factor (n = 342), ITD curves were folded into IPD curves (see 'Materials and methods'). The response at each IPD was treated as a vector with direction given by the IPD and length given by the mean firing rate at that IPD. The synchronization coefficient was the amplitude of the mean IPD vector divided by the sum of the mean firing rates for the entire period. Coefficients decreased minimally with frequency ($r^2 = 0.14$, $p < 0.001$) below 7 kHz. In sum, we found no evidence consistent with neural noise explaining the frequency-dependent ITD tuning. Thus, context dependence of the auditory cues resulting from the filtering properties of the HRTFs appeared to be the main source of location-dependent variability and the primary mechanism to adjust tonotopy along with spatial tuning.

# Discussion

In the present study we linked reliability of sensory cues with tuning properties of auditory neurons. Sounds at a given position do not yield identical localization cues over different contexts. However, for each location, there is a frequency range within which the localization cue IPD is most reliable. We showed that ICx neurons limit their frequency tuning to this range (*Figure 6*). Thus, the frequency tuning in the space-specific neurons of ICx is not simply due to tonotopy inherited from upstream neurons, but rather reflects a higher-order aspect of the neural code that may contribute to a more robust representation of sound location. Our study provides a case for how stimulus statistics can be captured by the neural processing.

Consistent with a trend reported in the optic tectum (*Knudsen, 1984*), we showed that the distribution of preferred frequency varies systematically with preferred ITD in ICx. This means that at each frequency, preferred ITDs are limited to a narrow part of the natural range. This is in stark contrast to the broad distribution of best ITDs across frequency reported in the upstream nucleus ICCc (*Wagner et al., 2002*, *2007*). Evidence for best ITDs covering the natural range in nucleus laminaris (NL) comes from studies of neurophonic potentials (*Sullivan and Konishi, 1986*) and axonal delays of input fibers to NL (*Carr and Konishi, 1990*). However, recent studies show that the methodology relying on the neurophonic to estimate the best ITDs of actual neurons, as opposed to the compound field potential from input fibers, may lead to spurious conclusions (*Mc Laughlin et al., 2010*; *Kuokkanen et al., 2013*). This suggests that the range of best ITDs at each frequency in NL deserves further investigation.

Selectivity for reliable localization cues could develop by Hebbian mechanisms favoring the least variable inputs. Indeed, the tuning of ICx cells adjusted to modifications of sound localization cues in juvenile and adult owls that had their facial ruff removed (*Knudsen et al., 1994*). Frequency-specific plasticity was also observed in owls raised with an acoustic filtering device that altered ITD in a frequency-dependent manner (*Gold and Knudsen, 2000*). When the device shifted the ITD by more than 50 µs, the ITD tuning of neurons also shifted by more than 50 µs compared to control owls. These studies indicate that ITD and frequency tunings can be adjusted by experience.

Frequency-dependent ITD tuning has been demonstrated in several other species and different brain regions, including the inferior colliculus of the guinea pig (*McAlpine et al., 2001*) and cat (*Hancock and Delgutte, 2004*; *Joris et al., 2006*), the medial superior olive of the gerbil (*Brand et al., 2002*; *Day and Semple, 2011*), and the lateral lemniscus of the chinchilla (*Bremen and Joris, 2013*). Thus, frequency-dependent ITD tuning is observed not only in cases where best ITDs fall largely out of the physiological range, such as in small rodents (*McAlpine et al., 2001*; *Brand et al., 2002*), but also in maps of auditory space where neurons are tuned to physiological ITDs, such as in the owl (*Wagner et al., 2007*). The correlation between ITD and frequency tuning thus represents a general organizational principle across species and coding schemes (*Schnupp and Carr, 2009*) for sound localization.

It has been proposed that owls and small mammals use different strategies to encode IPDs based on head size and the frequency range over which ITD can be encoded (*Harper and McAlpine, 2004*). The model of Harper and McAlpine (*Harper and McAlpine, 2004*) proposes that the optimal distribution of preferred IPDs at each frequency will maximize the information that the population provides about IPD. In this framework, the optimal distribution of preferred IPDs at each frequency depends on the statistical distribution of IPD from the environment. This theory predicts that preferred IPDs should cover the natural range of IPD at each frequency that is relevant for sound localization

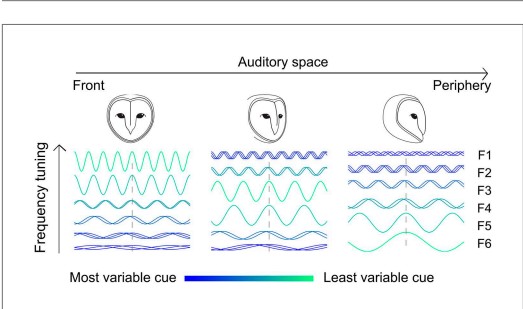

**Figure 6**. Adjusting tonotopy through weighting by reliability to represent space. Neurons receive inputs where different frequencies (F1–F6) are weighted by the IPD variance. For neurons tuned to frontal space (left), a larger weight is assigned to high frequencies where IPD is less variable, while neurons tuned to more peripheral space (middle and right) receive stronger input at lower frequencies. The effect of the head on IPD variability is indicated by the color (green is less variable) and superimposed sine waves (superimposed sinusoids shifted in phase indicate more variability).

in the owl. Harper and McAlpine took into account the statistics of the human acoustic cue to predict the optimal relationship between best IPD and frequency in humans. However, their predictions about the optimal neural code in the owl used a distribution of IPD that depends only on head size, and not on the IPD distribution occurring in natural environments. While this optimal coding model suggests that the distribution of IPD in natural environments may be an important factor influencing the neural code for sound localization, the model only addresses the representation of IPD at each frequency, and does not address how sound localization cues should be integrated over frequency to produce spatially selective auditory neurons. Here, we show that in the owl, spatial dependence of frequency tuning can be explained by a code that captures the range of frequency that carries the most reliable IPD. This study sheds light on an outstanding question in sound localization: Do neurons match the pattern of IPD across frequency experienced by each species? Our results suggest that neurons are tuned to the frequency range within which IPD varies least for each location over different environmental conditions. Thus, rather than neurons matching the IPD spectrum over the audible frequency range (**Brainard et al., 1992**; **Goodman et al., 2013**), they exclude ranges where the cue is most unreliable.

It has been suggested that the owl's brain represents the probability distribution of features in natural scenes (**Fischer and Pena, 2011**). Consistent with this idea is that the oblique effect in humans can be explained by a neural representation of the visual scene where vertical and horizontal orientations are more likely (**Girshick et al., 2011**). Our study further strengthens the idea that the brain represents the likelihood of natural features by showing that it synthesizes an internal model of stimulus reliability.

## Materials and methods

### Electrophysiology

The surgeries were performed as described previously (**Wang et al., 2012**). Briefly, two female adult barn owls were anesthetized with IM injections of ketamine hydrochloride (20 mg/kg; Ketaset) and xylazine (4 mg/kg; Anased). It has been shown that the responses of midbrain neurons are remarkably stable under ketamine anesthesia (**Ter-Mikaelian et al., 2007**; **Schumacher et al., 2011**). These procedures complied with National Institutes of Health and the Albert Einstein College of Medicine's Institute of Animal Studies guidelines. Responses were recorded with 1 MΩ tungsten electrodes (A–M Systems, Sequim, WA). Tucker Davis Technologies System 3 (Alachua, FL) and Matlab software were used to record neural data. ICx and ICCc neurons were identified by well-established physiological criteria based on their tuning to ITD, interaural level difference (ILD) and frequency, which permit unequivocal identification of recording sites (**Singheiser et al., 2012**).

### Sound stimulation

All experiments were performed inside a double-walled sound-attenuating chamber (Industrial Acoustics 120a, Bronx, NY) lined with echo-absorbing acoustical foam (Sonex 4″ wedge, Minneapolis, MN). These are rated to absorb sounds with highest efficiency down to below 1 kHz. Auditory stimuli delivered through calibrated earphones, consisting of a speaker (Knowles model 1914, Itasca, IL) and a microphone (Knowles model EK-23024) housed in a cylindrical metal earpiece and inserted in the owl's ear canal, consisted of 100 ms signals with a 5 ms rise-fall time at 10–20 dB above threshold. ITD tuning was initially estimated with broadband noise (0.5–10 kHz). ITD varied between ±300 μs in 30 μs steps over five trials. Frequency tuning was estimated with tones at the best ITD ranging between 600 Hz to 9000 kHz in 200 Hz steps, repeated over 15 to 20 trials. ITD tuning within the main peak of the rate-ITD curve was recorded at a finer resolution (10 μs steps; 20 repetitions) at the best ILD. The best ILD was determined as the ILD at the peak of the rate-ILD curve. ITD tuning to tonal stimulation was also measured. The range of ITD and sampling steps was adjusted by the stimulus frequency such that roughly three periods of the stimulating frequency were tested and 21 or more different ITDs were sampled. Each trial was repeated 20 times. Stimuli within all tested ranges were randomized during data collection.

To measure the effect of concurrent sounds on the spatial tuning of neurons in ICCc we used an array of 21 calibrated speakers placed inside a sound-attenuating chamber (**Wang et al., 2012**). The speaker array covered a range of ±100° in azimuth with an angular separation between speakers of 10°. To measure the spatial tuning with concurrent sound sources, we simultaneously played a broadband noise (0.5–10 kHz) at a given location while randomly activating other speakers of the array with

another broadband noise (within ± 90° from the preferred location in 20° steps). Dichotic and free-field recordings were performed in the same animals without disrupting the facial ruff.

## Data analysis

Recordings were performed at 138 ICx sites. Wave_clus was used for spike sorting (*Quiroga et al., 2004*). Briefly, spikes were detected using a voltage threshold set at five times the estimated standard deviation of the signal. To avoid double detection, spikes were separated by at least 1 ms. Neurons were considered isolated based on the presence of a refractory period of more than 1 ms in the inter-spike interval histogram and the similarity of spike shape. A complementary quality metric was the non-overlap of wavelet coefficients. Additionally, the results of the sorting algorithm were visually inspected to confirm the quality of the sorting. Consistent with previous reports (*Winkowski and Knudsen, 2006*), no significant differences were found between the results of sorted and non-sorted traces as neighboring ICx neurons have very similar tunings.

For each stimulus parameter, a rate curve was computed by averaging the firing rate across stimulus repetitions and subtracting the spontaneous rate. The best ITD was the ITD corresponding to the center of the main peak in the tuning curve. ITD tuning curves in ICx typically have a main peak and several smaller side peaks. We identified the main peak in the ITD tuning curve, then measured the ITD range where the firing rate was at least half the difference between the minimum and the maximum response. The best ITD was the center of this range of ITDs. We used the absolute value of the best ITD to combine data from contra- and ipsi-lateral sides as a function of the eccentricity of the receptive field.

For assessment of frequency tuning, the response area was defined as the frequency range that elicited more than 50% of the maximum response. The lower and upper edges of the frequency tuning curve were respectively the lowest and the highest frequencies of this range. Frequency tuning curves in ICx are broad and often lack an unequivocal peak. We thus defined the BF as the frequency at the center between the lower and upper edges. We also calculated the BF with a threshold at 30% and 75% of the maximum response or as the center of mass of the frequency tuning curve. No significant differences were found compared to the BF estimated with a threshold at 50% of the maximum response.

The gain (in dB SPL) was computed as the average of the left and right monaural gains. The monaural gains at each frequency represented the relative attenuation of the sound level by the HRTFs at each ear.

To compare the lower- and upper- edges of the frequency tunings with IPD reliability and gain, we computed the lower- and upper edges of the IPD reliability and gain. The lower- and upper- edges of the IPD reliability and gain were the lowest and the highest frequencies of the range that elicited more than 50% of the maximum IPD reliability and gain, respectively.

To quantify the neural variability across frequency, we calculated the Fano factors across trials in ITD curves with tonal stimulation. The Fano factor of an ITD curve is the ratio of the variance to the mean of the spike count.

We measured the strength of IPD tuning using circular statistics (*Goldberg and Brown, 1969*). The range of ITD used for this analysis was adjusted by the stimulus frequency, such that the number of data points relative to the period of the stimulus frequency was constant. We converted ITD to IPD by folding ITD curves from tonal stimulation into a single period of the stimulating frequency. IPD curves were fitted by a Gaussian function to achieve a uniform sampling of IPD over one period (*Perez and Pena, 2006*). The average response at each IPD was treated as a vector and the goodness of IPD tuning was estimated by a synchronization coefficient (*Goldberg and Brown, 1969*; *Kuwada et al., 1987*). The synchronization coefficient varies from 0 to 1, indicating no selectivity to IPD or perfect phase synchrony, respectively.

We reported the p-value and $r^2$ of the linear regression for estimating the relationships between variables. Pearson correlation coefficients were used to compare the smoothed frequency tuning curves with the estimated frequency from the IPD reliability and gain. The Pearson correlation coefficients of the spatial tuning curves obtained with single sources and those measured with concurrent sounds were averaged to quantify the overall effect of concurrent sounds in altering the spatial tuning of ICCc cells.

## Estimation of IPD variability

The HRTFs of ten barn owls were provided by Dr Keller (*Keller et al., 1998*). The gain of the loud-speaker used by *Keller et al., 1998* decreased sharply below about 2 kHz. In the present study, the

HRTFs were reprocessed by Dr Keller, equalizing the gain of the loudspeaker down to 1 kHz. As in the original paper, in order to remove the effects of the loudspeaker, an inverse filter for the loudspeaker was constructed and convolved with the head-related impulse responses. The new inverse filter, now optimized to include frequencies down to 1 kHz, removes most of the effects of the loudspeaker. To align HRTFs across owls, IPD variability was computed at an elevation centered between the acoustic axes of left and right ear-canals (the locations of maximum intensity gain of the HRTFs for left and right ears). This accounts for slight differences in head placement of each owl that might lead to differences in the definition of zero elevation across owls. We calculated the left and right acoustic axes by taking the median gain across the different frequencies and determined the center of gravity of the top 90% elevations.

To compute IPD variability for concurrent sound sources, broadband noise signals with flat spectra between 0.5 and 9 kHz and equal gain were each convolved with head-related impulse-responses at the appropriate source location. The outputs were passed through a gammatone filter-bank with center frequencies ranging from 1 kHz to 8 kHz in 0.2 kHz steps. The time constants of the filters were specific to the owl and estimated from *Koppl (1997a)* to model cochlear filters, as described in *Fischer et al. (2009)*. Because variability was estimated within narrow frequency bands, we used IPD, rather than ITD, by converting time into phase. The IPD in narrow frequency channels was calculated as the phase delay with the highest value of the cross-correlation between the left and right outputs of the gammatone filter. For each target location, we obtained 37 estimates of IPD, each with a second source located at one of the locations covering the frontal hemisphere (between −90 and 90° in steps of 5° at elevation zero). The circular standard deviation of IPD over locations of the second sound source was used as the estimate of IPD variability at each location.

To test whether reliability weighting allows ICx neurons to integrate coherent inputs in each frequency channel, we compared the average interaural correlation across locations of concurrent sounds with the IPD reliability. Concurrent broadband noise signals with flat spectra between 0.5 and 9 kHz and equal gain were convolved with head-related impulse-responses at the appropriate source locations. The outputs were passed through a gammatone filter-bank with center frequencies ranging from 1 kHz to 8 kHz in 0.2 kHz steps as described above. The cross-correlation was computed within each frequency channel. For a single sound source the interaural correlation was close to 1 and decreased when a concurrent sound was added. The interaural correlations with concurrent sounds were averaged across different locations of concurrent sounds.

## Acknowledgements

We are grateful to Clifford Keller and Terry Takahashi for providing the data and analyses on HRTFs and for feedback. We thank Bjorn Christianson and Adam Kohn for comments on the manuscript. We also thank Davide Reato and Michael Beckert for assistance with illustrations.

## Additional information

### Funding

| Funder | Grant reference number | Author |
|---|---|---|
| National Institute on Deafness and Other Communication Disorders | DC007690 | Jose L Pena |

The funders had no role in study design, data collection and interpretation, or the decision to submit the work for publication.

### Author contributions

FC, Conception and design, Acquisition of data, Analysis and interpretation of data, Drafting or revising the article; BJF, JLP, Conception and design, Analysis and interpretation of data, Drafting or revising the article

### Ethics

Animal experimentation: This study was performed in accordance with the recommendations in the Guide for the Care and Use of Laboratory Animals of the National Institutes of Health. Animals were

handled according to approved institutional animal care and use committee protocol (#20140409) of the Albert Einstein College of Medicine. Surgery was performed under anesthesia and every effort was made to minimize discomfort. Albert Einstein College of Medicine is fully accredited by the Association of Assessment and Accreditation of Laboratory Animal Care. Owls were held under a Scientific Collecting Permit from the US Fish & Wildlife Service (#MB06168B-0).

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
