## [Decision Letter]

Thank you for sending your work entitled “Cue reliability drives frequency-dependent spatial tuning in the owl's midbrain” for consideration at *eLife*. Your article has been favorably evaluated by Eve Marder (Senior editor) and 3 reviewers, one of whom, Ronald L Calabrese, is a member of our Board of Reviewing Editors.

The Reviewing editor and the other reviewers discussed their comments before we reached this decision, and the Reviewing editor has assembled the following comments to help you prepare a revised submission.

The authors present very thorough physiological and acoustical analyses of the spatial and frequency tuning of neurons in nucleus ICx of the Barn Owl. Interaural time difference (ITD) is the primary cue to localize sound in horizontal space in these birds. ITD is encoded in the firing rate of neurons that detect interaural phase difference (IPD) in nucleus ICx. In these birds IPD is influenced by the filtering properties of the facial mask so that IPD is most reliable for a particular frequency band at a particular location. They show that frequency tuning is such that IPD is most reliable at each preferred location (IPD). These findings have very important implications for how sensory circuits are constructed to capture stimulus reliability. This paper should provoke wide interest among the readers of *eLife*.

While the reviewers all agree on the interest and importance of the work, there are three concerns that must be addressed in revision. These revisions should not require new experiments but will require changes to the analysis and presentation.

1) Single-unit and multi-unit data should not be combined together. The main experimental finding is that best frequency (BF) varies systematically with best ITD. Determining BF and best ITD from multi-units is extremely problematical because it is not guaranteed that the underlying neurons all have the same tuning. Any correlation between BF and best ITD needs to be assessed with single unit data. Multiunit data should be removed from the data set, or at the very least analyzed and reported separately in text and figures.

2) It is confusing why, in the context of concurrent sounds, the authors choose to focus their cue reliability analysis on peak IPD and not interaural correlation. First off, it is not clear how they compute IPD. Authors state “the IPD of the gammatone filter outputs...” Normally one would cross-correlate the left and right filter outputs, then define the IPD as the phase delay with the highest correlation value (i.e., peak IPD). I assume that is how the authors compute it. But it is already known that this peak IPD will be midway between sources, as they cited from the Keller & Takahashi paper. The firing rates of ICx neurons aren't sensitive to the location of the peak IPD, they're sensitive to the interaural correlation at the best ITD (2). The addition of a concurrent source will decrease the interaural correlation, which is the value that is actually modulating the neural response. Interaural correlation at a specified IPD can be interpreted as IPD reliability; single sources will yield interaural correlation values near 1, while those for concurrent sources will be lower. It seems that analyzing IPD reliability as correlation is more directly relatable to the firing of ICx neurons. The problem with only looking at IPD peaks is that they could have little spread, which in this paper's analysis would suggest good IPD reliability, but yet have quite low correlation values, so actually not reliable cues.

3) The paper suffers from the current presentation. Although the authors attempt to present their results clearly, it was necessary to re-read sections and go back-and-forth several times to understand the set-up and the arguments. The overall flow and organization of both the text and the figures can be significantly improved. We strongly suggest that the authors revamp their manuscript by expanding the introduction, lay out the problem much more clearly in the introduction, clearly define and explain all the terms used, lay out more clearly the logic of their arguments, expand the figure legends, and re-organize figure panels. If done, the paper will be more widely accessible, and a broader audience, more than just researchers who work in auditory coding, will be able to appreciate these very interesting results.

[Editors' note: further revisions were requested prior to acceptance, as described below.]

Thank you for sending your work entitled “Spatial cue reliability drives frequency tuning in the Barn Owl's midbrain” for consideration at *eLife*. Your article has been favorably evaluated by Eve Marder (Senior editor), Ronald L Calabrese (Reviewing editor), and 2 reviewers.

The manuscript has now been revised according to reviewer comments and the presentation is much improved and the analysis more convincing. There are still some substantive concerns that necessitate further revision but will not require re-review beyond the BRE.

1) The authors state that they revised the results to only contain single-unit data. They now state that data is included from 177 ICx neurons (presumably single-unit), whereas in their original manuscript there were only 138 neurons (presumably multi- and single-unit). How did their N increase if multiunits were weeded out?

2) The authors misunderstood the new analysis regarding interaural correlation suggested in the last review. We were simply suggesting that they try defining IPD reliability as the average interaural correlation across contexts (evaluated from the cross-correlation function at the IPD of the target source), instead of the inverse of the variance of peak IPD across contexts. You could then compare frequency tuning to a heat map of this interaural correlation-based IPD reliability, similar to the other cases in Figure 4. We don't believe their new interaural correlation analysis is useful. You take interaural correlation functions from each frequency channel, normalize them, average them across channels (weighted or unweighted), then define the peak of this average function as the interaural correlation. The averaging of correlation functions is questionable, and defining interaural correlation as the peak of this function instead of the value at the IPD of the target source seems incorrect.

We believe that inclusion of a correct analysis of interaural correlation would give a better mechanistic understanding of why frequency tuning matches IPD reliability because the firing rates of ICc neurons are directly modulated by interaural correlation (shifts in peak IPD indirectly modulate firing rate via a shift in interaural correlation). We do not wish to delay the publication of this paper unnecessarily so we offer two options for revision. First option: Since the results with peak IPD are strong on their own, you could simply delete the newly-added section on interaural correlation in the Results and Methods and eliminate the two sentences in the Discussion pertaining to it. Second option: Perform the analysis detailed above and include that analysis in the paper to make even a strong case in support of your hypothesis.

---

## [Author Response]

*1) Single-unit and multiunit data should not be combined together. The main experimental finding is that best frequency (BF) varies systematically with best ITD. Determining BF and best ITD from multiunits is extremely problematical because it is not guaranteed that the underlying neurons all have the same tuning. Any correlation between BF and best ITD needs to be assessed with single unit data. Multiunit data should be removed from the data set, or at the very least analyzed and reported separately in text and figures*.

Originally, we had combined single and multi-unit data because it is assumed that in the external nucleus of the inferior colliculus (ICx) neighboring neurons have very similar tuning. We had compared the tuning from multi-unit recordings to the tuning from sorted traces and they were not significantly different, as reported previously (56). To address the reviewers' concern, we have sorted all our traces and now only report results from single-unit data. We describe the parameters of the spike-sorting algorithm (43) and criteria for single-unit classification in the Methods section. We did not find a significant difference between the results using multi- and single units.

*2) It is confusing why, in the context of concurrent sounds, the authors choose to focus their cue reliability analysis on peak IPD and not interaural correlation*.

Our analysis seeks to estimate how robust the input to ICx neurons is across contexts. We are assuming that these inputs are tuned to IPD within a narrow frequency band, which is consistent with the available literature. We would argue that variability of the 'Peak IPD' more precisely describes reliability within narrow frequency band. We reason that although ICx neurons respond to interaural correlation, their narrowly-tuned inputs will be independently affected by context within its frequency band.

We fully agree that it is nevertheless critical that the inputs remain coherent across frequency channels. We have confirmed that this is true by quantifying the interaural correlation that results from using the reliability weighting of frequencies and find that it retains or even improves the correlation relative to a uniform integration across frequency. This new analysis is now described in the Methods and reported in the Results.

*First off, it is not clear how they compute IPD. Authors state “the IPD of the gammatone filter outputs...” Normally one would cross-correlate the left and right filter outputs, then define the IPD as the phase delay with the highest correlation value (i.e., peak IPD). I assume that is how the authors compute it*.

The reviewers are correct on how we calculated the IPD. We have clarified that in the Methods section.

*But it is already known that this peak IPD will be midway between sources, as they cited from the Keller & Takahashi paper*.

Correct: we are relying on Keller and Takahashi to support our claim that IPD can shift in the presence of another source. We would like to note that this analysis is performed on a narrow-band basis.

*The firing rates of ICx neurons aren't sensitive to the location of the peak IPD, they're sensitive to the interaural correlation at the best ITD (*[2]*). The addition of a concurrent source will decrease the interaural correlation, which is the value that is actually modulating the neural response*.

While ICx cells are sensitive to the correlation across all inputs, we are examining the causes of variability in individual frequency channels across contexts. Our analysis shows that the second sound will not have a uniform effect on correlation across frequency but it will disrupt the signal more or less strongly for each frequency band depending on its position. We argue, and seek to demonstrate in the paper, that the system is sensitive to this, and neurons adapt their tuning to reliability within each frequency band. As discussed above, we now show that the interaural correlation is maintained at a high level when signals are integrated according to the reliability.

*Interaural correlation at a specified IPD can be interpreted as IPD reliability; single sources will yield interaural correlation values near 1, while those for concurrent sources will be lower*.

We agree that interaural correlation at a specific IPD can be interpreted as IPD reliability. We would like to point out that here we are not only specifying IPD but also frequency. Thus interaural correlation will be brought down by a concurrent source but each frequency band may contribute differently to it.

*It seems that analyzing IPD reliability as correlation is more directly relatable to the firing of ICx neurons. The problem with only looking at IPD peaks is that they could have little spread, which in this paper's analysis would suggest good IPD reliability, but yet have quite low correlation values, so actually not reliable cues*.

In this paper we argue that it is IPD reliability that matters because it governs the narrowband inputs to ICx neurons. Inputs that are more reliably active across contexts will remain and those unreliable will see their weights reduced. The spread is little but, given the sensitivity of the system, sufficient to turn on and off inputs in a context- and frequency-dependent manner. As we mention above, we now show that the interaural correlation is maintained at a high level when signals are integrated according to the reliability.

*3) The paper suffers from the current presentation. Although the authors attempt to present their results clearly, it was necessary to re-read sections and go back-and-forth several times to understand the set-up and the arguments. The overall flow and organization of both the text and the figures can be significantly improved. We strongly suggest that the authors revamp their manuscript by expanding the introduction, lay out the problem much more clearly in the introduction, clearly define and explain all the terms used, lay out more clearly the logic of their arguments, expand the figure legends, and re-organize figure panels. If done, the paper will be more widely accessible, and a broader audience, more than just researchers who work in auditory coding, will be able to appreciate these very interesting results. Details suggestions for altering the presentation are to be found in the minor comments of all three reviewers*.

We have extensively rewritten the manuscript, starting by the title, according to the comments of reviewers #1 and #3.

We have expanded the Introduction to better lay out our questions. Specifically, we have defined more clearly the terminology (ITD, IPD and BF) and the relationship between them in the sound localization system of the barn owl. We have also modified Figure 1 to more clearly describe how the auditory input is broken down across frequency, converted into IPDs and collapsed into ITD. We expanded the definition of “IPD reliability” in the Introduction. The concept of sensory reliability has been defined in the literature as the signal to noise ratio (precision) of sensory cues ([14], Ma & Jazayeri 2014). In our manuscript, IPD reliability is the precision of the peak IPD at a given frequency and location over different environmental contexts. We have also included motivating sentences and comparisons with the visual system to give intuitive notions of the concept of sensory reliability and sensory cue. We now state explicitly that “cue” refers to IPD.

We have reorganized the panels of Figures 3 and 4 (now Figures 3, 4 and 5) as suggested by reviewers #1 and #3. We have also expanded the figure legends, especially Figure 1 and the new Figure 5.

We have clarified several points in the Results and Methods sections, as requested by reviewer #2.

[Editors' note: further revisions were requested prior to acceptance, as described below.]

*1) The authors state that they revised the results to only contain single-unit data. They now state that data is included from 177 ICx neurons (presumably single-unit), whereas in their original manuscript there were only 138 neurons (presumably multi- and single-unit)*. *How did their N increase if multiunits were weeded out?*

We recorded from 138 sites in the owl's ICx. All recordings were processed by spike-sorting software (43) that groups the spikes into clusters based on the similarity of their shapes. Of these 138 recordings, 99 were validated as single units and 39 as multi-unit. The reason why the number of units increased from 138 to 177 is because we used all the single units that the spike sorting analysis produced, which in 39 out of 138 recordings was more than one. In the first submission we had combined single and multi-unit data (n=138) because it is assumed that neighboring neurons in ICx have very similar tuning. When units are considered well-isolated by the software we now include them in the analysis. We do not find a scientific justification to select one over another unit in the same recording, once the software classifies them as putative single units. Thus all single units yielded by the analysis are included, leading to a total of 177. We have clarified that in the Results section.

The reviewers’ concern for using multi-units was that it could affect the correlations between best frequency (BF) and best ITD because the tuning of neigborhing neurons could be different. We have confirmed that the tuning of sorted units from the same site were not significantly different, as reported previously (56). Additionally, regardless of which sample was used to compute the correlation between BF and best ITD, the results were all very similar (see figure below)Author response image 1.

The sample with sorted single units was used for the figures showed in the manuscript. To ensure that our correlation analysis convinces the readers, we now report the correlations with the 3 different samples in the Results section.

*2) The authors misunderstood the new analysis regarding interaural correlation suggested in the last review. We were simply suggesting that they try defining IPD reliability as the average interaural correlation across contexts (evaluated from the cross-correlation function at the IPD of the target source), instead of the inverse of the variance of peak IPD across contexts. You could then compare frequency tuning to a heat map of this interaural correlation-based IPD reliability, similar to the other cases in*
Figure 4*. We don't believe their new interaural correlation analysis is useful. You take interaural correlation functions from each frequency channel, normalize them, average them across channels (weighted or unweighted), then define the peak of this average function as the interaural correlation. The averaging of correlation functions is questionable, and defining interaural correlation as the peak of this function instead of the value at the IPD of the target source seems incorrect*.

*We believe that inclusion of a correct analysis of interaural correlation would give a better mechanistic understanding of why frequency tuning matches IPD reliability because the firing rates of ICc neurons are directly modulated by interaural correlation (shifts in peak IPD indirectly modulate firing rate via a shift in interaural correlation). We do not wish to delay the publication of this paper unnecessarily so we offer two options for revision. First option: Since the results with peak IPD are strong on their own, you could simply delete the newly-added section on interaural correlation in the Results and Methods and eliminate the two sentences in the Discussion pertaining to it. Second option: Perform the analysis detailed above and include that analysis in the paper to make even a strong case in support of your hypothesis*.

We apologize for having misunderstood the comment in the previous letter. We had interpreted that the reviewers were asking about the coherence across frequency channels, not across contexts. We have now calculated the average interaural correlation across locations of concurrent sources and normalized it at each location similarly to the IPD reliability. Not surprisingly, the results are very similar to the IPD reliability calculated from the standard deviation of the peak IPD (r^2^=0.81, p<0001, see figure below). In fact, if the shift of the peak IPD is large, the peak of the cross-correlation will also shift considerably. Thus a large standard deviation of peak IPD will often correspond to a smaller value of the interaural correlation at the IPD of the single sourceAuthor response image 2.

Because the IPD reliability computed with the peak IPD and the average interaural correlation are so similar we do not show a figure of the interaural correlation in the manuscript. Yet, as suggested by the reviewers, we have added a new paragraph to report the results on the interaural correlation. We also added a paragraph in the Methods to explain how we calculated the interaural correlation.

According to the reviewers’ suggestion, we have deleted the previously-added section on interaural correlation in the Results and Methods and eliminated the two sentences in the Discussion pertaining to it.
